# Identifying a Deferiprone–Resveratrol Hybrid as an Effective Lipophilic Anti-Plasmodial Agent

**DOI:** 10.3390/molecules26134074

**Published:** 2021-07-03

**Authors:** Supawadee Maneekesorn, Hataichanok Chuljerm, Pimpisid Koonyosying, Chairat Uthaipibull, Yongmin Ma, Somdet Srichairatanakool

**Affiliations:** 1Department of Biochemistry, Faculty of Medicine, Chiang Mai University, Chiang Mai 50200, Thailand; maneekesorn@gmail.com (S.M.); pimpisid_m@hotmail.com (P.K.); 2Department of Medical Technology, School of Allied Health Sciences, Walailak University, Nakornsrithammarat 80160, Thailand; hataichanokjj@gmail.com; 3National Center for Genetic Engineering and Biotechnology (BIOTEC), National Science and Technology Development Agency (NSTDA), Pathumthani 12000, Thailand; chairat.u@tcels.or.th; 4School of Pharmaceutical and Chemical Engineering, Taizhou University, Taizhou 318000, China; mayongminuk@hotmail.com or

**Keywords:** 3-hydroxypyridin-4-one, iron chelator, malaria, *Plasmodium falciparum*, resveratrol

## Abstract

Malaria i a serious health problem caused by *Plasmodium* spp. that can be treated by an anti-folate pyrimethamine (PYR) drug. Deferiprone (DFP) is an oral iron chelator used for the treatment of iron overload and has been recognized for its potential anti-malarial activity. Deferiprone–resveratrol hybrids (DFP-RVT) have been synthesized to present therapeutic efficacy at a level which is superior to DFP. We have focused on determining the lipophilicity, toxicity and inhibitory effects on *P. falciparum* growth and the iron-chelating activity of labile iron pools (LIPs) by DFP-RVT. According to our findings, DFP-RVT was more lipophilic than DFP (*p* < 0.05) and nontoxic to blood mononuclear cells. Potency for the inhibition of *P. falciparum* was PYR > DFP-RVT > DFP in the 3D7 strain (IC_50_ = 0.05, 16.82 and 47.67 µM, respectively) and DFP-RVT > DFP > PYR in the K1 strain (IC_50_ = 13.38, 42.02 and 105.61 µM, respectively). The combined treatment of DFP-RVT with PYR additionally enhanced the PYR activity in both strains. DFP-RVT dose-dependently lowered LIP levels in PRBCs and was observed to be more effective than DFP at equal concentrations. Thus, the DFP-RVT hybrid should be considered a candidate as an adjuvant anti-malarial drug through the deprivation of cellular iron.

## 1. Introduction

Iron is a key transition element for the growth, development and overall health of the metabolism of living organisms, including malaria parasites. Malaria is a fatal tropical disease caused by the *Plasmodium* spp. protozoa and transmitted by the female *Anopheles* mosquito vector. Epidemiologically, infection with *P. falciparum* malaria is severe and can be lethal in endemic countries including Thailand. At present, anti-malarial drugs, such as pyrimethamine (PYR), chloroquine (CQ), artemisinin (ART), cycloguanil, quinidine, amodiaquine and mefloquine, are used for the treatment of malaria; however, the emergence of drug resistance has become a significant concern [1,2,3]. Thus, an iron chelator may be used as an adjunctive anti-malarial drug to inhibit malaria growth and infection by depriving parasites of the iron which is essential for their survival [4,5]. Artemisinin (ART) is a pro-drug for its anti-malarial activity. This function is activated via a reductive cleavage of the endoperoxide ring by heme-iron in order to generate carbon-centered radicals. It then reacts with susceptible groups of plasmodial proteins, causing cellular damage and killing parasites [6]. Hence, a co-treatment of ART with an iron-chelating agent may interfere with the anti-malarial activity of this drug. Nonetheless, many iron chelators, including desferrioxamine (DFO), alkylthiocarbamates, 8-hydroxyquinoline, hydroxypyridine-4-one derivatives, *N*,*N*′-bis (o-hydroxybenzyl) ethylenediamine-*N*,*N*′-diacetic acid, *N*,*N*′-ethylenebis(o-hydroxyphenylglycine), ferrichromes, 7-nitrobenz-2-oxa-1,3-diazole-DFO conjugate, defersirox (DFX) and daphnetin have been studied as anti-malarial drugs that can be administered at different stages of the life cycle of parasites [7,8,9,10,11,12,13,14,15,16,17,18,19,20].

Deferiprone (DFP) is an orally active bidentate iron chelator that has been clinically used for the treatment of iron overload in thalassemia patients. It has also been used in a number of experimental investigations for the treatment of malaria [21]. Although DFP exhibits inhibitory effects on plasmodium parasite growth in vitro, it does not exhibit inhibitory effects and/or inflict defects in vivo because of its low lipophilicity. Additionally, it has been associated with a number of other side effects [8,14]. Further efforts are needed to develop orally active iron-chelating agents that exhibit specific anti-malarial action. Importantly, novel chelators should possess a greater degree of lipophilicity (partition coefficient, K_part_) in order to facilitate membrane transit and high iron-binding affinity (pFe). This would enable them to constantly deprive the erythrocytic labile iron pools (LIPs) that are essential for efficient plasmodium metabolism [22]. In recent years, polymeric/conjugative iron chelators have received attention as anti-malarial agents that can exhibit a range of pharmaceutical properties via their original small molecule counterparts [23]. For instance, daphnetin has been shown to exhibit iron-chelating, antioxidant and anti-malarial activities [18]. Synthetic 1,2,4,5-tetraoxanes were found to be more stable and potent as anti-malarial agents than conventional 1,2,4-trioxolanes [24]. An oral chelator, namely, (S)3″-(HO)-desazadesferrithiocin-polyether [DADFT-PE], displayed a lower degree of inhibition concentration at 50% (IC_50_) against *P. falciparum* than DFP and DFO [10]. Similarly, another orally active deferiprone–chloroquine hybrid (DFP-CQ) displayed a lower IC_50_ value against *P. falciparum* than conventional DFP (IC_50_ = 2.4 and 67 μM, respectively) [25]. Recently, we have developed our novel iron chelator, 1-(*N*-acetyl-6-aminohexyl)-3-hydroxy-2-methylpyridin-4-one (CM1), to be more lipophilic and efficient in the inhibition of *P. falciparum* growth and in the reduction in LIP than DFP [11]. 

Herein, the deferiprone–resveratrol hybrid (DFP-RVT) has been synthesized through a combination of the iron-chelating moiety of DFP in conjunction with the antioxidant structure of resveratrol (RVT) by Ping Xuet al. [26]. The compound has been labeled with the chemical name 2-(3,5-dihydroxystyryl)-5-hydroxy-1-methylpyridin-4(1H)-one and the chemical formula C_14_H_14_BrNO_4_(MW = 340 g/mol). It is represented by a metal-chelating diol part on the DFP molecule and an anti-oxidative part on the phenolic hydroxyl group (Figure 1). Extraordinarily, DFP-RVT has exhibited a better inhibitory effect against Cu^2+^/Fe^3+^-induced Aβ_1–42_ aggregation than RVT and curcumin (IC_50_ = 10.72, 11.89 and 18.73 μM, respectively). It has also displayed more potent antioxidant activity than Trolox, but less potent activity than RVT (IC_50_ = 1.73, 3.89 and 0.76 μM, respectively). Furthermore, it displays a stronger metal-chelating activity than DFP (pFe(III) = 19.6 and 20.6, respectively) [26]. In the present study, we aimed to investigate the lipophilicity of the DFP-RVT compound, its inhibitory effect on *P. falciparum* (3D7 and K1 strains) growth, and its ability to reduce LIP in *P. falciparum*-infected RBC.

## 2. Results

### 2.1. Partition Coefficient

Partition coefficient (*P*) values of free iron chelators and iron-chelates were determined by using a 1-octanol/water system and abbreviated as P*_o/w_*. As shown in Figure 2a, optical density (OD) values measured at 280 nm for both DFP-RVT and DFP were determined to be directly proportional to their applicable concentrations. After partitioning with 1-octanol, OD values of the aqueous DFP-RVT were clearly decreased (*p* < 0.05), whereas those of the aqueous DFP were slightly decreased when compared with the original values. Similarly, OD values of [DFP-RVT]-Fe(III) and DFP-Fe(III) complexes were decreased after being partitioned with 1-octanol, for which a significant level of reduction was found in the [DFP-RVT]-Fe(III) complex (Figure 2b). 

*P_o_*_/*w*_ values of free DFP and the DFP-Fe(III) complex were found to be 0.18 ± 0.06 and 0.21 ± 0.11, respectively, which were lower than the corresponding values of free DFP-RVT and the [DFP-RVT]-Fe(III) complex (2.73 ± 0.53 and 4.04 ± 0.01, respectively). A higher P*_o_*_/*w*_ value of DFP-RVT and its iron complex was observed when compared to that of DFP and its iron complex (Table 1). Thus, lipophilic DFP-RVT can readily penetrate the cell membrane to chelate intracellular iron, whereas the iron bound DFP-RVT complex with a higher *P_o_*_/*w*_ value would be able to emerge from the cells more easily.

### 2.2. Toxicity of DFP-RVT in PBMCs

When human healthy PBMCs were treated with various concentrations of DFP-RVT (0–200 µM) for 24 and 48 h. Consequently, cell toxicity was not found throughout all concentrations and during both periods of time (cell viability > 80%) with the exception of the 200 μM DFP-RVT treatment at 48 h (Figure 3).

### 2.3. Inhibitory Effect of DFP-RVT on P. falciparum Growth

We investigated the monotherapy of DFP-RVT, DFP or PYR in *P. falciparum* (wild-type 3D7 and PYR-resistant K1 strains), for which the resistance of the K1 strain was found to be caused by amino acid mutations of 59Cys->Arg and 108Ser->Asn on the *P. falciparum* dihydrofolate reductase gene. According to a comparison of drug sensitivity versus the percentage parasitemia curves (Figure 4a), PYR is recognized as a DHFR inhibitor drug that is a potent anti-malarial agent (IC_50_ = 0.046 μM). Furthermore, DFP-RVT exhibited a greater dose-response inhibitory effect on *P. falciparum* (3D7 strain) growth than DFP at IC_50_ values of 16.82 and 47.67 μM, respectively (Table 2). DFP-RVT (IC_50_ = 13.98 μM) was found to be the most potent anti-malarial agent against *P. falciparum* (PYR-resistant K1 stain) followed by DFP (IC_50_ = 42.02 μM) and PYR (IC_50_ = 105.6 μM), respectively (Figure 4b and Table 2). Notably, our study has revealed that the IC_50_ value of PYR for the *P. falciparum* 3D7 strain was approximately 3500-fold less than that for the *P. falciparum* K1 strain, whereas the IC_50_ values of DFP-RVT and DFP were almost equal in comparisons made between the two strains. 

In the combination therapy, treatments with DFP-RVT (5–15 µM) and 25 nM PYR have revealed a significant level of enhancement of anti-malarial activity against the cultured *P. falciparum* 3D7 strain in a dose-dependent manner (Figure 5a). Similarly, DFP-RVT treatments combined with 75 μM PYR significantly enhanced the anti-malarial activity against *P. falciparum* (KI strain) growth in a dose-dependent manner (Figure 5b). However, the IC_50_ value of DFP-RVT was not significantly different in both the PYR-sensitive (3D7) and PYR-resistant (K1) strains. Moreover, the combined treatment of DFP-RVT at concentrations within a range of 10–15 µM, along with PYR at a concentration of 75 µM, significantly increased the anti-malarial ability. Our findings highlight the hypothesis that our novel chelator, DFP-RVT, could be an effective lipophilic anti-malarial drug that inhibits both *P. falciparum* PYR-sensitive 3D7 and resistant K1 strains. Fantastically, the compound was able to kill the PYR-resistant strain, whereas PYR could not. Most importantly, DFP-RVT exhibited potent anti-malarial activity by PYR in the inhibition of *P. falciparum* (3D7 and K1 strains) growth. 

### 2.4. Effect of DFP-RVT Treatment on LIP Levels in PRBCs

In the calcein quenching technique, high measured fluorescence intensity (MFI) indicates a low level of intracellular iron sources such as LIP. Here, DFP-RVT and DFP (25–200 µM each) treatments significantly increased the MFI values of *P. falciparum*-infected red blood cells (PRBCs) in a concentration-dependent manner. This would imply that there was less LIP content than before the treatments were initiated (Figure 6a). Considerably, DFP-RVT was found to be more effective than conventional DFP at equal concentrations, along with a reduction in LIP levels in the PRBCs, as indicated by changes in MFI (∆MFI) (Figure 6b)**.**

## 3. Discussion

Iron is essential for the growth, development and overall health of the metabolisms of fast-dividing plasmodium parasites. Iron levels are also associated with anti-malarial drug resistance in endemic areas. Consequently, iron chelators would be an alternative or adjunctive anti-malarial agent in respective mono- or combined-therapy to eliminate severe malarial infections. Previous studies have shown that iron chelators effectively inhibited malaria growth at different stages. This was possibly achieved by depleting iron in the host PRBCs [7,8,9,10,11]. In addition, many synthetic hydroxypyridin-4-one (HPO) derivatives have elucidated better physicochemical properties (e.g., hydrophobicity/lipophilicity) and intracellular biological activities. These would include iron mobilization, free-radical scavenging and the anti-proliferative effects of Fao cells when compared to DFP [27,28]. Currently, a novel synthetic bidentate amino alcohol-conjugated HPO chelator has been reported as being more lipophilic and efficient in terms of iron-binding affinity than the parent DFP (log *P_o_*_/*w*_ values for free ligands = 0.56 versus −0.77; log *P_o_*_/*w*_ values for [Fe-L_3_] = 0.58 versus −2.6; iron-mobilizing efficacy in rats = 24.8 versus 11.8%, respectively); nevertheless, this chelator has never been assessed for anti-malarial activity, either in vitro or in vivo [29].

RVT with the scientific names of 3,5,4′-trihydroxy-trans-stilbene or 5-[(E)-2-(4-hydroxyphenyl)ethen-1-yl]benzene-1,3-diol (MW = 228.25 g/mol) involve polyphenols’ stilbenoids with multiple activities [30]. Herein, our novel iron chelator, DFP-RVT, exhibited a higher partition coefficient value (*P_o_*_/*w*_) than DFP. This would indicate that DFP-RVT possesses a greater degree of lipophilicity than DFP and, therefore, could penetrate the cell membrane faster. In most cases, this is regarded as true. Molecules such as DFP-RVT are soluble in solvents (water in this case) due to the interaction of their donor atoms (mainly oxygen atoms of DFP-RVT) with water protons via hydrogen bonds. This is because DFP-RVT is a bidentate iron chelator that needs three molecules to bind with one Fe to form the iron complex. Whenever the donor atoms of the DFP-RVT ligand are involved in the coordination bond with the metal, there is less possibility of interaction with the solvent. Therefore, the metal complex is usually more lipophilic than the ligand alone. This would suggest that DFP-RVT is a good iron chelator. The stoichiometric binding of iron to the DFP-RVT hybrid was previously determined using ultraviolet–visible spectrophotometry. The ratio of iron:hybrid = 1:3 was suggested by Xu et al. [26]. Likewise, DFP-RVT showed a higher partition coefficient value when it was bound with iron and formed an iron-chelate, Fe(III)-[DFP-RVT]_3_, which meant that the iron complex of DFP-RVT could efflux from the cell more easily than the free ligand. Apparently, the ability to penetrate the cells of DFP-RVT directly correlated with the efficacy of cellular iron chelation. Accordingly, DFP-RVT reduced cytoplasmic/lysosomal LIP levels in PRBCs to a greater degree than DFP at equal concentrations. These findings suggest that even after applying equal concentrations, the higher LIP-depleting action of DFP-RVT could merely be the effect of it having accumulated to a greater degree inside the cells because it is much more lipophilic. It may also be possible that DFP-RVT and DFP would have access to different forms of LIP or intracellular iron. Accordingly, this would be a point of particular interest for future investigations. DFP-RVT contains a cluster of two hydroxyl phenyl groups derived from RVT molecules linked to DFP molecules; therefore, the modification would successfully improve the lipophilicity and iron-chelating efficiency. Several studies have demonstrated the ability of the iron chelator to inhibit malaria growth with regard to the degree of lipophilicity for free and bound forms by penetrating the biomembrane and exhibiting iron-binding affinity [15,31,32]. In comparison with hydrophilic DFO (MW = 560.7 g/mol), the smaller and more lipophilic chelators, such as DFP and DFX (MW = 373.4 g/mole), were markedly effective in terms of gaining access to and chelating the intracellular LIP. Furthermore, they were consequently more effective at preventing the oxidative injury of cells [33]. Purportedly, an increase in iron chelator lipophilicity could improve or enhance the biological/pharmacological activities in the cells.

A previous study has reported that the plasmodium parasite utilized available labile iron pools in the cytoplasm of host red cells rather than the toxically oxidant hemozoin iron present in the food vacuoles for the synthesis of heme components in the mitochondria and apicoplast [34]. In one study, an orally active DADFT-PE, coded as FBS0701, was investigated and was found to exhibit a more potent inhibitory effect against *P. falciparum* than DFP and DFO (IC_50_ = 6, 15 and 30 μM, respectively) [10]. Pattanapanyasat and colleagues reported that HPO derivative chelators efficiently exhibited anti-malarial activity in conjunction with the inhibition of *P. falciparum* in both PYR-sensitive and resistant strains [35]. In addition, the DFP-CQ hybrid exhibited a more potent inhibition of *P. falciparum* growth than DFP (IC_50_ = 2.4 and 67 μM, respectively) [25]. Moreover, CM1, which exhibited more lipophilicity than DFP ((K_part_ = 0.53 and 0.17, respectively), was more efficient in the inhibition of *P. falciparum* growth (IC_50_ = 35.14 and 58.25 μM, respectively) and in the reduction in LIP than DFP [11]. Undoubtedly, DFP-RVT more efficiently inhibits *P. falciparum* growth than DFP, as exhibited by the lower IC_50_ values (at the micromolar level) when compared with DFP. However, the IC_50_ values of DFP-RVT and DFP on PYR-sensitive and resistant *P. falciparum* protozoa were not determined to be significantly different. The iron chelator could kill the malaria parasite by the deprivation of essential iron in the cytoplasm of PRBCs, or may act directly with the functional iron-containing components and enzymes in the plasmodium parasites [36]. Furthermore, the combination treatment exhibited additive effects between DFP-RVT and PYR, in which DFP-RVT enhanced the activity of PYR to inhibit *P. falciparum* growth in both 3D7 and K1 strains. DFP showed a degree of non-toxicity to the normal cell line, in which IC_50_ was higher than 200 µM [37]. Consistently, DFP-RVT treatments at 25–200 μM were not found to be toxic to humans at normal PBMC levels and at 24 h of incubation, but were slightly toxic at 200 μM at 48 h of incubation. Due to the emerging and increasing strains of drug-resistant malaria, combinations of anti-malarial drugs with oral iron chelators may present a more efficient form of anti-malarial chemotherapy. Consistently, combinations of anti-malarial drugs (e.g., CQ and PYR) with iron-chelating agents (e.g., DFO) resulted in an additive degree of inhibiting *P. falciparum* (FCR-3 strain) growth in vitro [38]. In contrast, supplementation with iron in children with uncomplicated *P. falciparum* malaria, who received PYR or CQ, improved their anemic condition without increasing their susceptibility to malaria [39]. However, the treatment of patients diagnosed with malaria-associated anemia, who were administered with an anti-malarial drug together with oral iron, exhibited significant increases in blood hemoglobin levels, but failed to clear blood parasitemia when compared to patients who received the anti-malarial drug treatment alone [40]. Under careful consideration, anemic *P. falciparum*-infected patients receiving iron chelation therapy did not experience any effects in terms of erythropoietic activities such as hemoglobin synthesis. Overall, the most favorable properties of the novel deferiprone-resveratrol hybrid, in terms of its therapeutic efficiency against *P. falciparum* infection and the inherent non-cytotoxic hybrid, were observed. These properties probably relate to the optimal lipophilicity of the hybrid molecule, its ability to penetrate the PRBC membrane, and the chelation of cytoplasmic/lysosomal none-heme iron that can inhibit *P. falciparum* growth in both anti-malarial chemotherapeutics. Furthermore, the co-operation of iron chelators and pyrimethamine will be of significant interest in future investigations for their potential to inhibit plasmodium growth and development.

## 4. Materials and Methods

### 4.1. Chemicals and Reagents

Dimethyl sulfoxide (DMSO)(density 1.10 g/mL) was purchased from Santa Cruz Biotechnology, Inc. (Dallas, TX, USA), whereas 1,2-dimethyl-3-hydroxypyridin-4-one, DFP or GPO-L-One^®^ (MW = 139), was kindly donated by the Institute of Research and Development, Government Pharmaceutical Organization, Bangkok, Thailand. Additionally, 1-octanol (MW = 130.23, CAS-No 111–87–5), Wright–Giemsa staining solution, lysis buffer, fetal bovine serum (FBS) and pyrimethamine (PYR, MW = 248.7) were purchased from Sigma-Aldrich Chemicals Company (St. Louis, MO, USA). Calcein acetoxymethyl (CA-AM, 1 mg/mL in DMSO, Catalogue Number C3099), 3-(4,5-dimethylthiazolyl-2)-2,5-diphenyltetrazolium bromide (MTT, Catalogue Number M6494), SYBR Green I nucleic acid gel-stain (10,000× concentrate in DMSO, Catalogue Number S7563) and SYTO^®^61 red-fluorescent nucleic acid dye (Catalogue Number S11343) were purchased from Invitrogen, Molecular Probes (Thermo Fisher Scientific, Waltham, MA, USA). RPMI-1640 (Gibco^®^Invitrogen) incomplete medium and phosphate-buffered saline (PBS) were purchased from Life Technologies Corporation (Carlsbad, CA, USA.)

### 4.2. Methods

#### 4.2.1. Synthesis of DFP-RVT

DFP-RVT was synthesized by Dr. Yongmin Ma and colleagues from Zhejiang Chinese Medical University, Hangzhou, China. The compound has been labeled with the chemical name 2-(3,5-dihydroxystyryl)-5-hydroxy-1-methylpyridin-4(1H)-one and the chemical formula C_14_H_14_BrNO_4_ (MW = 340 g/mol). The design, synthesis and biological evaluation of a series of DFP-RVT compounds were described by Ping Xuet al. [26] with regard to their potent iron-binding and anti-oxidative properties.

#### 4.2.2. Determination of Partition Coefficient

To study the hydrophilicity or lipophilicity of the compounds, measurements of the partition coefficient (*P_o_*_/*w*_) value were based on the 1-octanol/water system [41]. In this study, 1-octanol was mixed with the free ligand of iron chelators (i.e., DFP-RVT and DFP) and their complexes with iron at concentrations in a range of 0–100 µM. After that, the mixture was stirred at room temperature for 45 min and allowed to stand for 15 min. The aqueous phase was transferred to a quartz cuvette and measured in terms of the OD at 280 nm for the free ligands, 450 nm for the iron complex of DFP and 375 nm for the iron complex of DFP-RVT. Accordingly, *P_o/w_* values were calculated according to the following formula:*P_o/w_* = (OD_1_ − OD_2_)/OD_2_ × *V_w_*/*V_o_*(1)
where OD_1_ and OD_2_ represent the absorbance values of the aqueous phase solution before and after adding 1-octanol, respectively; *V_w_* and *V_o_* represent the volumes of aqueous and 1-octanol phases, respectively. In this study, a ratio of *V_w_*/*V_o_* =1 was reported [41]. 

#### 4.2.3. Human Ethics 

Blood collection was permitted by the Director of Maharaj Nakorn Chiang Mai, Faculty of Medicine, Chiang Mai University, Chiang Mai, Thailand (Reference Number 8393.8.9/436). The study protocol was approved by the Research Ethics Committee for Human Study, Faculty of Medicine, Chiang Mai University, Chiang Mai, Thailand (Research ID 7575, Study Code: BIO-2563/Date of Approval, 28 September 2020).

#### 4.2.4. Toxicity Study in Peripheral Blood Mononuclear Cells

A colorimetric MTT assay was used to assess the viability of live cells with mitochondrial active reductase enzymes [42]. Venous blood was collected from healthy human volunteers, transferred to a sodium heparin tube, and an equal volume of 0.85% NaCl solution was added. LymphoPrep™ density gradient solution was added to the heparinized whole blood (at a ratio of 1:8, *v*/*v*) and centrifuged at 900× *g* for 30 min. Peripheral blood mononuclear cells (PBMCs) layered at the interphase were then collected, washed twice with RPMI 1640 medium, and maintained in RPMI 1640 containing 10% FBS [43]. In the assay, cells (5 × 10^3^ cells/well, viability > 80%) were loaded into culture plates, treated with DFP-RVT (0–200 µM) at 37 °C for 24 and 48 h, and an MTT solution was added. After incubation, blue-colored formazan product occurring in the cells was extracted with DMSO and OD was photometrically measured at 540 nm and 630 nm. 

#### 4.2.5. *P. falciparum* Culture

*P. falciparum* (PYR-sensitive 3D7 and resistant K1 strains)-infected RBCs (PRBCs) were cultured in human blood group O+ erythrocytes with 4% hematocrit (Hct), RPMI 1640 medium supplemented with 0.3 g/L L-glutamine, 5 g/L hypoxanthine and 10% pooled human serum incubated in a 37 °C incubator with 5 % CO_2_ for 48 h (one cycle of *P. falciparum*) [11]. A thin-film PRBC smear was stained with Wright–Giemsa dye solution, and the numbers of PRBCs were counted to a total of one hundred RBCs. PRBCs were then maintained maximally at 10–15% parasitemia in the culture medium. 

#### 4.2.6. Sorbitol Synchronization of *P. falciparum*-Infected RBCs

Ring-stage (not later than 10–12 h post infection) PRBC suspension was spun down at 1800× *g* at room temperature for 3 min and the supernatant was discarded. The cell pellets were resuspended in 5 % (*w*/*v*) sterile D-sorbitol solution (10 mL) and incubated at 37 °C for 10–15 min. After centrifugation, cell pellets containing only the intraerythrocytic ring- and early trophozoite-stage parasites were resuspended in a new culture medium to obtain an RBC suspension (4% Hct) and used in the experiments [44].

#### 4.2.7. Drug Sensitivity Test for *P. falciparum*

The efficacy of the potential iron chelator DFP-RVT in *P. falciparum* 3D7 and K1 strains was assessed and compared to the reference anti-malarial drug PYR with regard to their IC_50_ values. In cases involving numerous blood samples, an SYBR Green I fluorescent probe was used to stain plasmodium DNA in PRBCs, which was then detected using flow cytometry [45]. In brief, DFP-RVT and DFP (0–200 µM each), and PYR (0–200 nM and 0–200 µM for the PYR-sensitive and resistance strains, respectively) were prepared in 1% DMSO solution. Drug solutions (1 μL) were added to the synchronized ring-stage PRBCs (1% parasitemia and 2% Hct) in a black 96-well plate (100 µL) and incubated under 5% CO_2_ atmospheric conditions for 48 h. Finally, 100 µL of SYBR Green I solution, which was freshly prepared in the lysis buffer (20 mM Tris-HCl, 5 mM EDTA, 0.008% saponin and 0.08% Triton X) (SYBR Green I:lysis buffer = 0.2 µL:1 mL), was added to each well and incubated in the dark at room temperature for 1 h. MFI was then determined at excitation and emission wavelengths of 485 and 530 nm, respectively [45,46,47]. Parasite survival curves were made by plotting drug concentrations (log_common_ scale) on the x-axis versus the percentage of parasitemia on the y-axis, whereas the IC_50_ value was determined from each curve. In the combined treatment, *P. falciparum* 3D7 and K1 strain-infected RBCs were treated with DFP-RVT (0–15 µM) in the presence of 25 nM and 75 µM PYR, respectively, for 48 h under 5% CO_2_ atmospheric conditions. After incubation, the treated PRBCs were stained with SYBR Green I solution, FI was measured using flow cytometry, and a survival curve was constructed according to the method previously described.

#### 4.2.8. Measurement of LIP in *P. falciparum*-Infected RBC

Erythrocyte LIP levels were stoichiometrically detected with fluorescent calcein derived from esterase hydrolysis of a non-fluorescent CA-AM ester using flow cytometry [11], in which an increase in measured FI was related to a decrease in LIP. In brief, *P. falciparum*-infected RBC suspension (5% trophozoite stage and 0.2% Hct) was incubated with 0.25 µM CA-AM at 37 °C for 15 min, washed twice with RPMI-1640, and resuspended in incomplete medium. Cell suspension was then inoculated into 96-well plates and treated with or without an iron chelator, DFP or DFP-RVT (0–200 µM). After that, SYTO-61 dye (1 µM) was added to the treated cells, and they were incubated at 37 °C for 1 h under a standard culture system. Finally, FI signals (wavelengths of 485/530 nm for calcein and wavelengths of 625/645 nm for SYTO 61) of the cells were analyzed using a flow cytometer (Model BD FACSAria™ III Cell Sorter, BD Biosciences, San Jose, CA, USA). Changes in the LIP levels observed from ∆MFI with or without treatment were calculated by employing the following formula:∆MFI = MFI_CA-AM/ironchelator_ − MFI_CA-AM/alone_(2)
where ∆MFI ≤ 0 indicates LIP availability, and ∆MFI > 0 indicates LIP depletion.

#### 4.2.9. Statistical Analysis

Data are expressed as mean ± SEM or mean ± SD values. Statistical significance was determined using the analysis of variance (ANOVA) or multiple tests, for which a *p*-value < 0.05 was considered significant.

## 5. Conclusion

Improvements in the hydrophobicity of DFP after being hybridized with resveratrol have clear benefits fort the plasma membrane permeability of anti-malaria drugs. Consequently, a novel DFP-RVT hybrid has been found to be more lipophilic and efficient in reducing the amount of LIP in parasitized red blood cells when compared with DFP. Accordingly, DFP-RVT inhibited malaria growth and enhanced therapeutic activity more efficiently than the parent DFP and reference PYR in both mono- and combined-therapies. Moreover, DFP-RVT was determined to be nontoxic to PBMCs at concentrations up to 200 µM. In further studies, DFP-RVT will be investigated in *P. berghei*-infected mice with regard to its bioavailability and anti-malarial efficacy. Furthermore, hepatic glucuronidation and the oxidative metabolism of the hybrid need to be intensively investigated in animals.

## Figures and Tables

**Figure 1 molecules-26-04074-f001:**
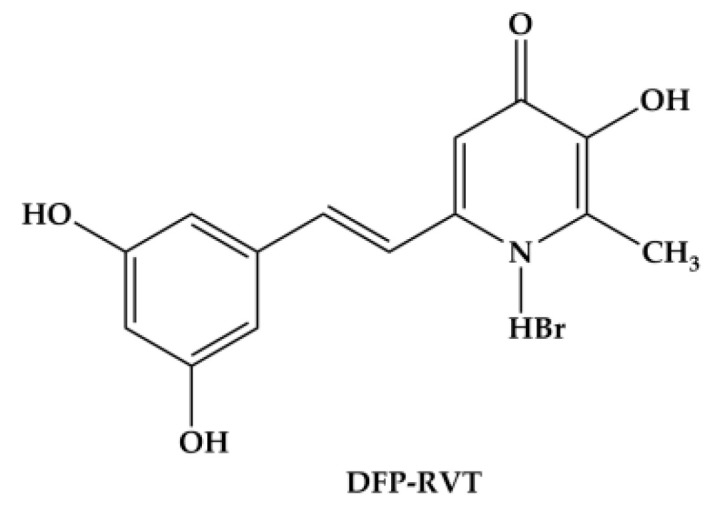
Chemical structure of the deferiprone-resveratrol hybrid (DFP-RVT) (redrawn from [26]).

**Figure 2 molecules-26-04074-f002:**
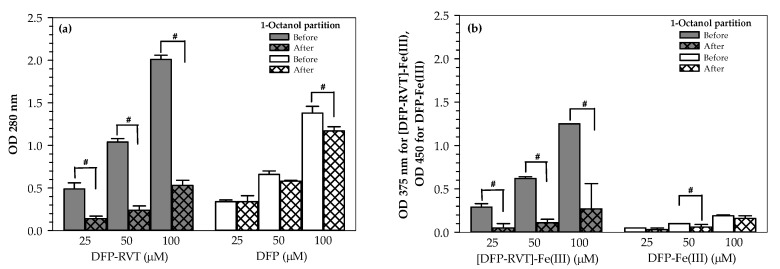
Partition of DFP-RVT, DFP, [DFP-RVT]-Fe(III) complexes and the DFP-Fe(III) complex in a water/octanol solvent. (**a**) DFP and DFP-RVT (0–100 µM each) solutions were partitioned in 1-octanol and centrifuged, while their aqueous phases were measured in terms of optical density (OD) at 280 nm using spectrophotometry. (**b**) DFP-RVT and DFP (0–100 µM each) solutions were incubated with ferric ammonium citrate solution and then partitioned in 1-octanol and centrifuged, while their aqueous phases were measured in terms of OD at 375 and 450 nm, respectively, using spectrophotometry. OD values obtained from three repetitions are expressed as mean ± standard deviation (SD) values. Statistical significance was determined using an analysis of variance test, for which a *p*-value of <0.05 was considered significant. ^#^
*p* < 0.05 when compared before and after the addition of 1-octanol.

**Figure 3 molecules-26-04074-f003:**
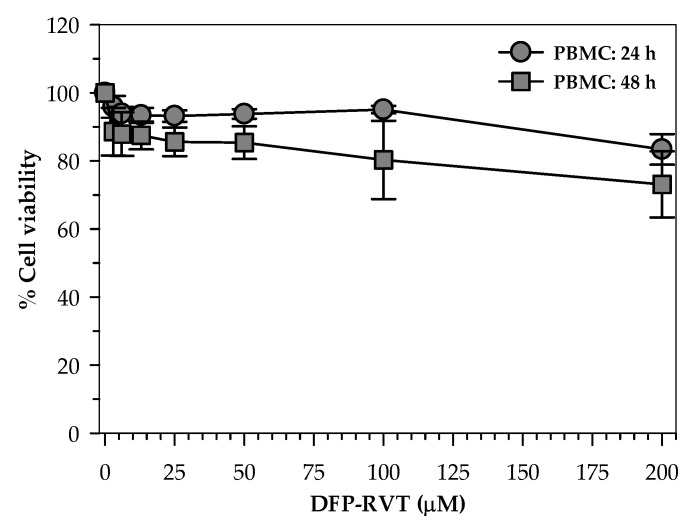
Percent viability of healthy human PBMCs treated with DFP-RVT for 24 and 48 h. Data obtained from three independent triplicate experiments are expressed as mean ± SD values.

**Figure 4 molecules-26-04074-f004:**
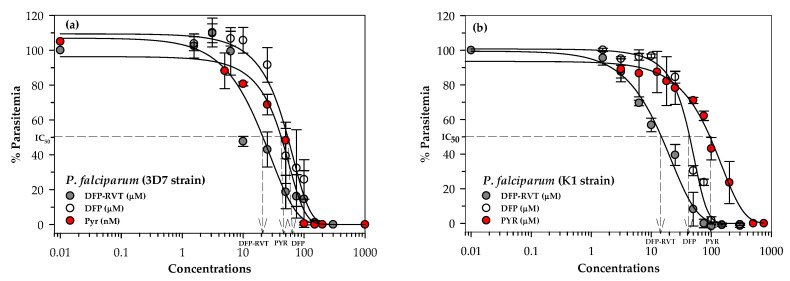
Inhibitory effects of DFP-RVT, DFP and PYR monotherapy on the growth of the *P. falciparum* 3D7 strain (**a**) and the K1 strain (**b**). Data obtained from three independent triplicate experiments are expressed as mean ± SD values.

**Figure 5 molecules-26-04074-f005:**
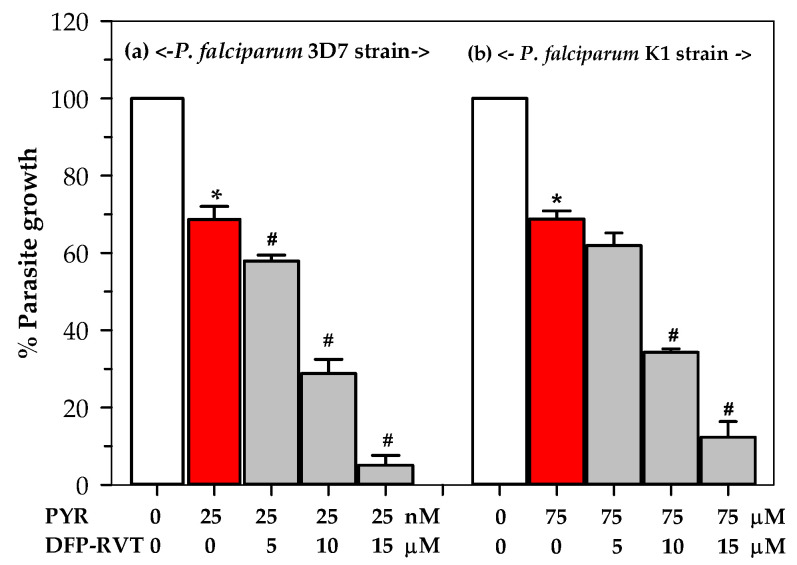
Inhibitory effects of combined PYR and DFP-RVT treatments on the growth of *P. falciparum* 3D7 strain (**a**) and K1 strain (**b**). Data obtained from three repetitive experiments are expressed as mean ± SD values. * *p* < 0.05 when compared without treatment; ^#^
*p* < 0.05 when compared with 25 nM PYR treatment for the 3D7 strain and 75 μM for the K1 strain alone.

**Figure 6 molecules-26-04074-f006:**
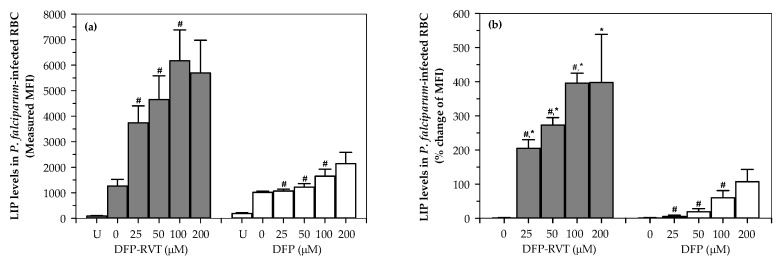
Intracellular LIP levels in *P. falciparum* (3D3 strain)-infected RBCs treated with DFP and DFP-RVT (0–200 µM each). Change in mean fluorescence intensity (ΔMFI) (**a**) and percentage changes of the MFI (**b**) values obtained from three independent experiments are presented as mean ± standard error of the mean (SEM) values. Statistical significance was determined using multiple *t*-tests, for which a *p*-value of <0.05 was considered significant. ^#^
*p* < 0.05 when compared with consecutive lower concentrations; * *p* < 0.05 when compared with DFP at equal concentrations.

**Table 1 molecules-26-04074-t001:** Partition coefficient (*P_o_*_/*w*_) values obtained from three repetitions are expressed as mean ± SD values. * *p* < 0.05 when compared with DFP; ^*f*^ *p* < 0.05 when compared with the free ligands. DFP: deferiprone; RVT: resveratrol; DFP-RVT: deferiprone-resveratrol hybrid.

Compounds	*P_o_*_/*w*_ Values
DFP	0.18 ± 0.06
DFP-iron complex	0.21 ± 0.11
DFP-RVT	2.73 ± 0.53 *
[DFP-RVT]-iron complex	4.04 ± 0.01 *^,^^*f*^

**Table 2 molecules-26-04074-t002:** Absolute IC_50_ values of DFP-RVT, DFP and PYR for inhibition of *P. falciparum* growth.

Compound	Absolute IC_50_ Value (µM)	
	*P. falciparum* (3D7 Strain)	*P. falciparum* (K1 Strain)
DFP-RVT	16.82	13.98
DFP	47.67	42.02
PYR	0.05	105.61

## Data Availability

The data presented in this study are available on request from the corresponding author.

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
