# Peer review of "Identifying a Deferiprone–Resveratrol Hybrid as an Effective Lipophilic Anti-Plasmodial Agent"

_molecules, 2021, doi:10.3390/molecules26134074_

Round 1
Reviewer 1 Report
The work by Maneekesorn and colleagues describes the application of a dual-activity molecule (iron chelator and antioxidant resveratrol, DFP-RVT) in the treatment of Plasmodium infection both in wild type and pyrimethamine-resistant strains.
DFP-RVT was nontoxic to human host cells, more toxic than parent DFP and also more hydrophobic, which confers good pharmacological potential. Also, it depleted labile iron pools from the parasite, which is theorized as the basis of its antimicrobial activity.
The paper is well written and based. I have only minor comments to make.
Lines 42-43: I did not understand the statement “This activity requires heme iron to break its epoxide ring”. Which epoxide ring? And what that has to do with interfering with artemisinine activity? Please consider revision.
Line 103 and throughout the text: all the “micro” symbols are changed by squares here and throughout the text. Please revise.
Line 114: I believe I can offer a possible explanation for the higher lipophilicity of the iron complex in comparison to the parent chelator. Molecules are soluble in solvents (water in this case) by the interaction of its donor atoms (mainly oxygens of DFP-RVT) with water protons via hydrogen bonds. Whenever the donor atoms of the ligand (DFP-RVT) are involved in the coordination bond with the metal, there is less possibility of interaction with the solvent, therefore the metal complex is usually more lipophilic than the ligand alone.
Lines 126 and 127: Is it PBMC or PRBC? This is confusing throughout the text
Lines 182-183 and 215-216: There is I believe a caveat in the comparison of the LIP depletion caused by DFP-RVT versus DFP. The comparison assumes that equal EXTRAcellular concentrations of the ligands will translate into equal INTRAcellular concentrations of the ligands, but this has not been demonstrated. Therefore, the higher LIP-depleting action of DFP-RVT could be merely the effect of it being more accumulated inside the cells – after all, it is much more lipophilic. I believe that this should be remarked. (another possibility would be that DFP-RVT and DFP have access to different forms of LIP or intracellular iron; this could be an interesting point for a future investigation)
Line 212: Was the stoichiometry of the complex [Fe(DFP-RVT)3] determined here or in the original publication? If it was here, I didn’t see how it was done.
Reviewer 2 Report
The manuscript entitled “Identifying Deferiprone-Resveratrol Hybrid as an Effective Lipophilic Iron-Chelating Agent to Kill Plasmodium falciparum” is about to evaluate the potential of Deferiprone-Resveratrol Hybrid as an antimalarial agent.
The manuscript written well and experiments designed well but need some modifications to be published in this journal.
Typing error at line 39: “theemergence”
At figure 2, the concentration is not correctly presented.
At the toxicity of DFP-RVT, it is not clear what the PBMC or PRBC mean. I found what does PRBC mean at the line 180; infected red blood cells (PRBC), so authors need full name for abbreviation at the first time showed in this manuscript.
It is not clear whether the IC50 value of PYR against DHFR is obtained by this experiments or referenced from previous paper. I mean it is not clearly mentioned that some of data were actually obtained by experiments or just referenced. So I recommend authors make sure how to obtain the results and if referenced indicate the reference paper.
At line 169, PRY should be a PYR.
At figure 5, please check the concentration of compounds.
Generally check the spacing.
At last, I am not convince about the title, because within entire manuscript, I could not see the iron-chelating activity of DFP-RVT. The iron-chelating activity of DFP-RVT supposed to be reported before and I think this paper focused on the antimalarial activity. So I recommend that authors should change the title that focus to killing activity of DFP-RVT.
Reviewer 3 Report
Submitted manuscript Molecules-1267220 “Identifying Deferiprone-Resveratrol Hybrid as an Effective Lipophilic Iron-Chelating Agent to Kill Plasmodium falciparum” is a nice piece of work and well written. Please keep an eye on the following errors.
Title of the manuscript should change “Kill Plasmodium falciparum” does not look good. Incorporate the word like ex vivo and anti-plasmodial.
In figure 4 write down the unit of concentration. In other figures as well micro (µM) or nM is missing. Take care of these mistake.
Statistical tests and platform/software used for analysis should be mentioned in the methods section.
Similarly, some other minor mistakes are also there including spelling and English. I will advise to the author/s that review and revise the manuscript at your end.
Overall, this paper needs some proofreading and rearrangements according to the reader point of view.
All the best.
